# Analysis of Functional Domains in Rho5, the Yeast Homolog of Human Rac1 GTPase, in Oxidative Stress Response

**DOI:** 10.3390/ijms20225550

**Published:** 2019-11-07

**Authors:** Carolin Sterk, Lauren Gräber, Hans-Peter Schmitz, Jürgen J. Heinisch

**Affiliations:** AG Genetik, Fachbereich Biologie/Chemie, Universität Osnabrück, Barbarastr. 11, 49076 Osnabrück, Germany; Carolin.Sterk@biologie.uni-osnabrueck.de (C.S.); graeber.lauren@web.de (L.G.); hans-peter.schmitz@biologie.uni-osnabrueck.de (H.-P.S.)

**Keywords:** *Saccharomyces cerevisiae*, Rho-GTPases, mitochondria, GFP trapping, signalling

## Abstract

The small GTPase Rho5 of *Saccharomyces cerevisiae* is required for proper regulation of different signaling pathways, which includes the response to cell wall, osmotic, nutrient, and oxidative stress. We here show that proper in vivo function and intracellular distribution of Rho5 depends on its hypervariable region at the carboxyterminal end, which includes the CAAX box for lipid modification, a preceding polybasic region (PBR) carrying a serine residue, and a 98 amino acid–specific insertion only present in Rho5 of *S. cerevisiae* but not in its human homolog Rac1. Results from trapping GFP-Rho5 variants to the mitochondrial surface suggest that the GTPase needs to be activated at the plasma membrane prior to its translocation to mitochondria in order to fulfil its role in oxidative stress response. These findings are supported by heterologous expression of a codon-optimized human *RAC1* gene, which can only complement a yeast *rho5* deletion in a chimeric fusion with *RHO5* sequences that restore the correct spatiotemporal distribution of the encoded protein.

## 1. Introduction

The necessity to react to changing environmental conditions has led to the establishment of complex but largely conserved signal transduction pathways in all eukaryotic cells from yeast to humans. Many of those pathways use monomeric G-proteins of the Ras superfamily as molecular switches, which are active in their GTP-bound and inactive in their GDP-bound state and serve as central and highly conserved components [1]. To promote activation, guanine nucleotide exchange factors (GEFs) trigger the dissociation of GDP and its replacement by the intracellularly prevailing GTP. GTPase activating proteins (GAPs) stimulate hydrolysis of GTP and thus inactivation of the G-protein [2]. Frequently, guanine nucleotide dissociation inhibitors (GDIs) stabilize the GDP-bound state by preventing the association with GEFs and simultaneously sequester the GTPase from its membrane compartment [3]. The importance of the Ras-type GTPases is reflected in a variety of diseases associated with their malfunctions, ranging from developmental disorders and modulation of the immune system to neurodegenerative diseases (for further reading, see references in studies [4,5]).

The ras homology (Rho) subfamily of these GTPases in the model yeast *Saccharomyces cerevisiae* is represented by six members, namely Rho1 to Rho5 and Cdc42, which regulate signaling pathways involved in polarized growth, exocytosis and cytokinesis [6]. While the roles of the two essential members, Rho1 in cell wall integrity (CWI) and Cdc42 in establishment of cell polarity under different physiological conditions, have been thoroughly investigated, the diverse biological functions of Rho5 in yeast, and its Rac1 homologs in other fungi and humans, are only beginning to emerge [7,8]. Deletions of the gene in haploid *S. cerevisiae* strains were first reported to be hyper-resistant to cell wall stress agents and suggested a negative regulatory role for Rho5 in cell wall integrity (CWI) signaling [9]. Moreover, hyper-activation of Rho5 resulted in an increased sensitivity to salt stress, linking its function to the high osmolarity glycerol (HOG) signaling pathway [10]. In addition, the observation that *rho5* mutants were hyper-resistant to oxidative stress provided a link to apoptosis [11] and autophagy/mitophagy, which also involves CWI and HOG signaling [12,13].

We recently showed that Rho5 interacts with a dimeric GEF constituted of yeast Dck1 and Lmo1, with deletion mutants in either of the encoding genes resembling *rho5* deletions in their hyper-resistance toward oxidative stress–induced cell death [14]. All three components of the putative trimeric Dck1-Lmo1-Rho5 (DLR) complex rapidly translocate from the cell periphery to mitochondria under such stress conditions, followed by mitophagy and cell death [14]. The translocation of Rho5 occurs only in the presence of both Dck1 and Lmo1 and is also triggered by glucose starvation, thus linking the DLR complex to nutrient signaling [15].

An essential characteristic of GTPases to fulfil their biological functions in different subcellular compartments is their association with membranes, which is primarily achieved by a modification of the C-terminally located CAAX-motif (i.e., a cysteine residue followed by two aliphatic amino acids and a variable C-terminal residue). Processing of this motif results in removal of the three terminal amino acids, a carboxymethylation, and geranyl-geranylation of the cysteine, altogether providing a hydrophobic anchor for association with target membranes [16]. A polybasic region (PBR) preceding the CAAX motif is also present in many Rho-type GTPases, where the positive charges electrostatically interact with the negatively charged membrane phospholipids to enhance membrane association and specificity [17,18]. Together with some preceding amino acid residues, the two motifs have been designated as a hyper-variable region, which is involved in many interactions with effector proteins and the spatiotemporal distribution [17].

Yeast Rho5 and its human homolog Rac1 share the basic features of Rho-GTPases, reflected in the conservation of their primary sequences (Figure 1A). Thus, the P-loop and the two switch regions involved in nucleotide binding are highly conserved from yeast to humans. The P-loop contains a glycine residue substituted by a valine in an oncogenic, constitutively active human Ras variant (G12V; [19]), and hyperactivation of yeast Rho5 is also observed in the respective mutant as well as the Q91H substitution within the switch II region [9]. The CAAX box in both Rho5 and Rac1 is preceded by a PBR region, which only in the yeast protein carries a serine residue. Rho5 also differs from Rac1 by a 98 amino acid insertion prior to the PBR region, as well as a shorter insertion of 30 amino acids between the switch I and switch II domains (Figure 1A). The hypervariable region at the C-terminal end of human Rac1 has been suggested to have a major influence on its intracellular distribution and physiological targets [17], and we reasoned that this is where important regulatory properties of the yeast homolog should be encoded, and may be altered by genetic manipulations to study the phenotypic outcome. We therefore set out to investigate the structural requirements that govern the translocation of Rho5 to mitochondria under oxidative stress, other than its obligatory association with the dimeric GEF. In this context, we focused on the role of conserved primary sequence features, especially those located in the C-terminal part of the GTPase, and their role in response to oxidative stress. Knowledge gathered from these experiments was used to construct a functional Rac1-Rho5 hybrid protein that complements a yeast *rho5* deletion and may be a valuable tool to study human Rac1 functions.

## 2. Results

### 2.1. The Carboxyterminal CAAX Motif of Rho5 is Required for Plasma Membrane Association and Physiological Function

Sequence alignments revealed a strong conservation between yeast Rho5 and human Rac1, with some specific features unique to the yeast GTPase (Figure 1A). In order to study the physiological significance of the Rho5 domains, we started by investigating the importance of the conserved C-terminus for its physiological function and intracellular distribution related to oxidative stress. A schematic summary of the mutant alleles studied in this work is presented in Figure 1B. First, we exchanged the cysteine at position 328 for a leucine to eliminate the presumed prenylation and carboxymethylation site and compared this mutant to wild type, a complete *rho5* deletion, and a strain expressing the constitutively active *RHO5*^G12V^ allele. To avoid phenotypic variations in growth assays, which may be caused by fluctuating plasmid copy numbers, the wild-type gene and the mutant alleles were introduced at the native *RHO5* locus using *SkHIS3* as a selection marker inserted 3′ to the coding sequence. Growth was monitored in rich medium under standard growth conditions and under oxidative stress. Confirming previous observations, the *rho5* deletion was hyper-resistant toward hydrogen peroxide, whereas the constitutively active variant *RHO5*^G12V^ was hyper-sensitive as compared to wild type (Figure 2A). As expected, neither allele impaired growth in the absence of oxidative stress. Next, we used the previously observed synthetic lethality of *rho5* and *sch9* deletions to test the in vivo function of our mutant alleles [15]. Whereas the *rho5* deletion did not produce any viable progeny in a *sch9* background, the constitutively active *RHO5^G12V^* variant produced small colonies typical of *RHO5 sch9* segregants (Figure 2B). On the other hand, the *rho5^C328L^* mutant, which lacks the cysteine residue required for lipid modification, resembled the characteristics of the *rho5* deletion both in growth on rich media with and without oxidative stress and in its synthetic lethality in combination with a *sch9* deletion (Figure 2A,B). We concluded that the GTPase is not functional in the absence of lipid modification.

To follow the intracellular distribution, *CEN/ARS* vectors encoding GFP-Rho5 fusions were introduced into a *rho5* deletion strain, which also carried an mCherry mitochondrial marker construct. A strain with the wild-type GFP-Rho5 fusion showed the expected plasma membrane (PM) localization under standard growth conditions and the rapid translocation to mitochondria under oxidative stress, as reported previously (Figure 2C; [14]). Likewise, the activated GFP-Rho5^G12V^ resided predominantly at the PM under standard growth conditions but displayed an incoherent distribution within different cells of the same culture upon addition of hydrogen peroxide, varying from mainly mitochondrial (41%), through cytosolic (48%), to some PM-associated (11%; Figure 2C). On the other hand, a GFP-tagged Rho5^T17N^ mutant, which cannot be activated, did not interfere with mitochondrial localization, since 73% of the cells showed colocalization of the GFP signal with the mitochondrial marker under oxidative stress (Appendix A).

As expected from the lack of a lipid modification, GFP-Rho5^C328L^ entirely failed to localize to the PM in life-cell fluorescence microscopy but appeared rather uniformly distributed within the cytosol under standard growth conditions (Figure 2C). A substantial portion of cells still showed translocation of the GFP signal to mitochondria upon exposure to oxidative stress (36% as compared to 70% of cells with the wild-type GFP-Rho5 fusion). Thus, although causing a significant reduction, lipid modification is not absolutely required for mitochondrial translocation.

### 2.2. Rho5 Requires the Basic PBR Region for Proper Plasma Membrane Association and Physiological Function

To study the function of the PBR sequence preceding the CAAX-box, we first constructed a mutant in which the six lysine residues were replaced by alanines (Rho5^pbrK6A^; Figure 1B). Again, the respective allele was introduced at the native *RHO5* locus using *SkHIS3* as a selection marker inserted 3′ to the coding sequence. The *RHO5^pbrK6A^* strain grew like wild type under standard conditions but showed an increased resistance to hydrogen peroxide, although not reaching that of a *rho5* deletion strain (Figure 3A). This suggested a partial functionality of this mutant with regard to oxidative stress response. Likewise, a synthetic growth defect was observed in crosses with the *sch9* deletion (Figure 3B). Thus, although segregants containing the mutant *rho5^pbrK6A^* allele were viable in the *sch9* background, they produced much smaller colonies than those carrying *RHO5 sch9*. Similar to the CAAX-box mutant, GFP-Rho5^pbrK6A^ failed to associate with the PM under standard growth conditions, but upon exposure to hydrogen peroxide still translocated to mitochondria in 32% of the cells observed (Figure 3C). We also noticed that GFP-Rho5^pbrK6A^ was not dispersed evenly within the cytosol, but frequently associated with ring-like structures, which we could not yet identify, but they were clearly different from mitochondria.

A distinguishing feature of *S. cerevisiae* Rho5 as compared to human Rac1 is the presence of a serine residue within the PBR, which has been shown to modulate membrane association of another human Ras-type GTPase depending on its phosphorylation state [20]. Site-specific mutants carrying either an alanine residue that cannot be phosphorylated (Rho5^S326A^) or a phosphomimetic glutamate substitution (Rho5^S326E^) at this position both produced a functional Rho5, as judged from their growth curves in the presence of hydrogen peroxide (Figure 3A) and their viability in a *sch9* background (Figure 3B). While strains producing Rho5^S326A^ behaved basically like wild-type in the growth experiments, while strains with Rho5^S326E^ showed a slight increase in H_2_O_2_-resistance (Figure 3A). GFP-fusions of the GTPase with either the serine or the glutamate substitution localized to the PM under normal growth conditions very similar to wild type. They also translocated predominantly to mitochondria under oxidative stress, but with a larger fraction remaining at the PM as compared to the wild type (Figure 3C). Mitochondrial translocation under stress appeared to be somewhat reduced for GFP-Rho5^S326E^ (58%) as compared to the 70% of both GFP-Rho5 (Figure 2C) and GFP-Rho5^S326A^ (Figure 3C).

Taken together, these results indicate that the basic character of the PBR region is required for proper PM association of Rho5 and influences the association of Rho5 with mitochondria under oxidative stress, whereas the serine residue appears to be of minor importance.

### 2.3. The C-Terminal Extension is Required for Rho5 Function in Yeast

A distinctive feature of *S. cerevisiae* Rho5 versus human Rac1 is the presence of a 98 amino acid extension between residues P221 and D320 prior to the PBR-CAAX regions (Figure 1A), which does not resemble any protein domains available in the databases, and whose functional significance has not yet been studied. Thus, we removed this sequence (with the resultant protein designated Rho5^∆222–319^) and tested the response of the respective mutant to oxidative stress. Compared to the wild-type, *rho5^∆222–319^* cells displayed the hyper-resistance typical of a complete *rho5* deletion and were synthetically lethal with a *sch9* deletion, indicating that the GTPase is not functional in vivo (Figure 4A,B). Interestingly, a GFP-Rho5^∆222–319^ fusion did not associate with the PM under normal growth conditions and translocation to mitochondria was drastically reduced under oxidative stress (Figure 4C). Since this GFP-Rho5^∆222–319^ fusion resembled the mutants in the CAAX motif and the pbrK6A variant in their failure to associate with the PM, the minimal requirement of the Rho5 C-terminus for such an association became an issue. In order to address this question, we introduced a number of different GFP fusion constructs (compare Figure 1B) and determined their intracellular distribution.

In order to pinpoint the requirements for PM association, we first demonstrated that a GFP fusion with the C-terminal 110 amino acids of Rho5 (GFP-Rho5^C110^), which comprises the entire extension terminated by the PBR/CAAX motifs, correctly localizes to the PM under standard growth conditions (Figure 4D). It largely remains at that position after addition of hydrogen peroxide, with only a small fraction translocating to mitochondria. As expected from the importance of the basic PBR sequence demonstrated above, substitution of the lysines by alanine residues (K6A) impaired PM localization of this construct. The Rho5 sequence was then further truncated to include only the last 17 amino acids in the GFP fusion (GFP-Rho5^C17^), leaving three more basic residues prior to the PBR sequence. This fusion localized at the PM in the absence of oxidative stress, whereas translocation to mitochondria was virtually absent after application of hydrogen peroxide, with the majority of the signal observed in the cytosol and non-mitochondrial internal structures (Figure 4D).

These observations prompted us to revisit the GFP-Rho5^∆222–319^ fusion, which completely lacked PM localization. We introduced the five amino acids preceding the PBR sequence present in the GFP-Rho5^C17^ construct, to obtain GFP-Rho5^∆222–314^. Indeed, with this variant GFP signals were predominantly found at the PM under normal growth conditions and the fusion protein behaved very similar to GFP-Rho5^C17^ under oxidative stress, with the majority of signals detected in the cytosol and internal structures. However, a higher percentage of cells displayed colocalization with mitochondria for GFP-Rho5^∆222–314^ under these conditions (Figure 4C). Nevertheless, substitution of the wild-type *RHO5* allele by the one encoding Rho5^∆222–314^ neither reduced the hyper-resistance of the cells toward hydrogen peroxide in growth experiments (Figure 4A) nor compensated the synthetic lethality with a *sch9* deletion (Figure 4B), indicating that the additional five amino acids cannot restore the in vivo function of the internally truncated GTPase.

In summary, while the C-terminal 17 amino acids of Rho5 are sufficient to direct the protein to the PM, neither these nor the entire C-terminal 110 amino acids confer an efficient translocation to mitochondria under oxidative stress.

### 2.4. In Vivo Trapping of GFP-Rho5 to the Mitochondrial Surface Affects Its Physiological Functions

To further address the question of the physiological significance of Rho5 localization, a GFP trap tethering the GTPase to the mitochondrial surface was employed. To construct this mitochondrial GFP-trap, we fused the GFP-binder sequence to the N-terminus of the transmembrane domain of the outer mitochondrial protein Fis1 on a yeast integrative plasmid and inserted it at the *leu2-3,112* locus. As expected from the presence of a wild-type *FIS1* allele in these strains, segregants carrying this construct did not display any growth defects in tetrad analyses. Such a segregant was crossed with a strain carrying a GFP gene fusion at the *RHO5* locus to obtain haploid strains with combinations of *GFP-RHO5, FIS1^TMD^-GB* and the respective wild-type loci. Fluorescence microscopy demonstrated that the traps worked as expected, i.e., they either recruited GFP-Rho5 exclusively to mitochondria in the *GFP-RHO5, FIS1^TMD^-GB* segregants, both in the absence and presence of hydrogen peroxide (Figure 5A). Growth curves showed that segregants with separate *GFP-RHO5* and *FIS1^TMD^-GB* alleles reacted similar to wild type to hydrogen peroxide (Figure 5B). On the other hand, the strain expressing both *GFP-RHO5* and *FIS1^TMD^-GB* resembled the hyper-resistance observed for the *rho5* deletion, demonstrating that the GTPase is not functional when trapped exclusively to mitochondria. We reasoned that this phenotype could be attributed to a necessity for Rho5 to be activated at the PM prior to its translocation in order to fulfil its function in oxidative stress response. Therefore, we introduced an activated *GFP-RHO5^G12V^* allele at the native *RHO5* locus and combined it with a *FIS1^TMD^-GB* background by crossing and tetrad analysis. Segregants with GFP-Rho5^G12V^ trapped to the mitochondria displayed wild-type sensitivity toward hydrogen peroxide demonstrating in vivo functionality of the GTPase, while strains only producing GFP-Rho5^G12V^ without the trapping device proved to be hyper-sensitive, as observed above for the non-tagged version (Figure 5C).

### 2.5. A C-Terminally Modified Human Rac1 Partially Complements a rho5 Deletion

Since impairment of human *RAC1* functions is associated with a variety of diseases, we decided to explore the possibility of using *S. cerevisiae* as a heterologous host for its expression and molecular analysis. For this purpose, a codon-optimized *RAC1* sequence was cloned into a *CEN/ARS* vector and expressed either under the control of the native *RHO5* promoter or that of the strong glycolytic *PFK2* promoter. Neither construct complemented the *rho5* deletion, as judged from the retained hyper-resistance toward oxidative stress (Figure 6A). In order to rule out that the lack of complementation was due to a failure of Rac1 to be activated by the yeast GEF Dck1/Lmo1, a constitutively active *RAC1^G12V^* allele was also expressed, but did not complement the *rho5* defects, either.

Based on the functional significance of the C-terminal insertion of 98 amino acids for the yeast GTPase described above, we created a chimeric construct, in which coding regions for the PBR and CAAX motifs of *RAC1* were replaced by the C-terminal sequences of *RHO5* (encoding Rac1-Rho5^C110^). Upon overexpression in a *rho5* deletion, this construct restored the sensitivity toward hydrogen peroxide even more than that of the wild type, demonstrating its functionality in yeast (Figure 6A). Complementation capacity was also confirmed by viability of strains carrying this plasmid in a *rho5 sch9* double deletion background (Figure 6B). It should be noted that, although growth could be restored in these transformants, colonies of the viable segregants were smaller than those carrying only the *sch9* deletion with a *RHO5* wild-type allele.

GFP fusion constructs showed that while the unmodified GFP-Rac1 did not localize at the PM or mitochondria both in the absence and in the presence of hydrogen peroxide, GFP-Rac1-Rho5^C110^ appeared almost exclusively at the PM under normal growth conditions and a substantial portion translocated to mitochondria under oxidative stress (Figure 6C). Finally, substitution of just the Rac1 PBR and CAAX motifs for yeast Rho5^C17^ conferred PM localization to the respective GFP fusion under standard growth conditions but failed to translocate to mitochondria under oxidative stress (Figure 6C).

## 3. Discussion

Although Rho5 has been identified as the last of the six yeast Rho-family members more than 15 years ago, then as a negative regulator of CWI signaling, its exact mode of action in various physiological processes is not yet fully understood [8]. Neither is the role of its protein domains, which comprise evolutionary conserved sequences as well as features specific for *S. cerevisiae* Rho5. The mutational analyses presented herein provide first experimental evidence for the functional significance of these domains in vivo.

For Rac1, a human homolog of yeast Rho5, a puzzling variety of physiological functions has been reported, which to a large extent may be mediated by what was defined as a hyper-variable region (HVR) at the C-terminal end of Rho-type GTPases (reviewed in reference [17]). The HVR comprises both the CAAX box with the cysteine residue as a target for lipid modification and the polybasic region (PBR), which are both conserved in yeast Rho5 (Figure 1), as well as some preceding amino acid residues. Substitution of the CAAX-box cysteine at position 189 in Rac1 for a serine abolished both prenylation and in vivo function [21]. Likewise, our Rho5^C328L^ variant proved to be non-functional in two different physiological read-outs, i.e., the hyper-resistance against hydrogen peroxide typical for complete *rho5* deletions as seen in growth curves (oxidative stress response; [14]), and the synthetic lethality in combination with a *sch9* deletion (carbon stress response; [15]). In vivo fluorescence of the GFP-tagged Rho5^C328L^ demonstrated that it does no longer associate with the PM under standard growth conditions, in contrast to the wild-type version. Nevertheless, the rapid translocation of GFP-Rho5 to mitochondria upon application of oxidative stress is not completely abrogated in the prenylation mutant. While this indicates that some interaction with the dimeric GEF Dck1/Lmo1 necessary for mitochondrial association is still possible, the moiety reaching the mitochondria is clearly not sufficiently active to mediate the oxidative stress response. Moreover, this provided a first hint that the GTPase has to be activated at the PM prior to its translocation in order to fulfil its physiological function. We presume that the less efficient mitochondrial translocation of the activated Rho5^G12V^ variant is due to a reduced affinity toward the dimeric GEF, since the latter should interact primarily with the non-activated conformation of the GTPase. Accordingly, the non-activatable Rho5^T17N^ translocated like wild type to mitochondria under oxidative stress, indicating that its interaction with Dck1/Lmo1 is not impaired.

PM association may also be aided by electrostatic interactions of the basic residues within the PBR with membrane phospholipids, depending on the combination of their arginine and lysine residues [22]. We here showed that the PBR of Rho5 is necessary but not sufficient for proper membrane targeting. More importantly, quantitative trapping of GFP-Rho5, but not that of its activated GFP-Rho5^G12V^ variant, resulted in a loss of function in the oxidative stress response. This further supports the notion that the GTPase needs to be activated at the PM prior to its translocation to mitochondria, in line with the general assumption that activation of Rho-type GTPases occurs while they are associated with membranes [23]. For Rho5, this activation would be mediated at the PM by its dimeric GEF Dck1/Lmo1, which is also required for its translocation to mitochondria [14]. This is reminiscent of the situation in mammalian cells, where insulin-induced activation of Rac1 is mediated by interaction with its GEF at the PM [24]. It is also consistent with other data on Rac1, where prenylation of the cysteine residue was shown to be required for its activation and its subsequent interaction with target proteins, which included the response to oxidative stress [25].

For Rac1, the HVR not only mediates membrane association through the prenylated cysteine residue and the PBR, but also determines its specific interaction with effector proteins and its subcellular distribution, which frequently involves some preceding amino acid residues [17]. We here found that a yeast-specific insertion of 98 amino acids prior to the PBR is required, but not sufficient, to ensure the rapid translocation of Rho5 to mitochondria under oxidative stress and trigger the proper cellular response. Thus, the phenotypes of Rho5^Δ222–319^ and Rho5^Δ222–314^ mutants resemble those of the complete *rho5* deletion, demonstrating its functional importance. However, GFP-Rho5^C110^, which carries the entire insertion and the native C-terminal end, translocated poorly to mitochondria. This suggests that N-terminal sequences, presumably the switch regions interacting with the dimeric GEF Dck1/Lmo1, are also required for this process, as previously observed for interactions of Rac1 with some effectors [17]. We therefore propose that the yeast-specific insertion mediates the interaction of Rho5 with its effectors. Supporting this notion is the hydrophilic, probably surface-exposed nature of this sequence predicted by secondary structure analyses. In a recent paper, Singh and co-workers reported on a large number of putative Rho5 interactors, comprising, amongst others, the two eisosome components Lsp1 and Sur7 [26]. Lsp1 directs the membrane invaginations typical for eisosomes through its BAR domain [27]. Mammalian Rac1 was also found to interact with the SH3 domains of BAR domain proteins through its proline residues in the HVR, generating intracellular tubular structures [28]. While it is thus tempting to speculate that the tubular structures observed in some of our GFP-fusions with different Rho5 variants are caused by a dysregulation of its interaction with Lsp1, this remains to be determined.

Another interesting interaction was reported between mammalian Rac1 and a protein kinase C-related kinase (PRK1; [29]). PRKs are characterized by the presence of homology repeats, so-called HR1 domains, at their N-terminal end, which are also present in the unique, prototypic protein kinase C (Pkc1) of *S. cerevisiae* and interact with the Rho1-GTPase [30]. Pkc1 is a key component of the CWI pathway, to which Rho5 function was found to be related [9]. An evolutionary conserved interaction between Rho5 and Pkc1 would thus be an option, although it was neither detected in the work of Singh and co-workers [26] nor in direct two-hybrid interaction assays with the HR1 domains of Pkc1 [31]. Yet, since both assays were performed under standard growth conditions, it cannot be ruled out that an interaction might still occur in specific stress situations.

Mammalian PKC was found to phosphorylate K-Ras at a serine residue within its PBR, causing the dissociation of the GTPase from the PM and its association with mitochondria and other endomembranes [20]. Other GTPases of the Ras-superfamily were also found to be phosphorylated at serine residues within the PBR in order to modulate their interaction with downstream effectors [32,33,34,35,36]. In contrast, our data show that substitutions of serine 326 in the PBR of yeast Rho5 did neither drastically effect its distribution to different membranes, nor its physiological functions determined by the two different outreads. The minor increase in resistance of Rho5^S326E^ toward growth in the presence of hydrogen peroxide as compared to wild type could indeed be due to a weakened PM association caused by an electrostatic repulsion of the negative charge introduced. Although we thus cannot exclude the possibility that phosphorylation of serine 326 by Pkc1 or other kinases occurs in Rho5, we believe that it would be of minor regulatory importance, at least for the oxidative stress response and the reaction to carbon stress. Large-scale phosphoproteome studies performed under standard growth conditions further support this notion, since several serine residues appeared to be phosphorylated in Rho5 but not serine 326 [37].

A common feature of the HVR in mammalian GTPases is the presence of a nuclear localization sequence (NLS) within the PBR [18], which is also present in Rho5. In Rac1, the NLS and the preceding proline residues determine its cell-cycle dependent import into the nucleus [38]. The authors found that lipid modification of the CAAX-box cysteine impairs this function, since cysteine-substitution mutants showed a much stronger nuclear localization. In contrast, yeast GFP-tagged Rho5^C328L^ showed no sign of nuclear localization in the fluorescence images, and neither did the GFP fusion to the C-terminal 17 amino acids. We thus believe that a nuclear function of yeast Rho5 is highly unlikely.

In summary, the data discussed above show that a correct spatiotemporal distribution of Rho5, its activation either exclusively or predominantly at the PM, and its specific interactions with downstream effectors work hand in hand to ensure its physiological functions. Last but not least, these finding also enabled us to construct a hybrid human Rac1-Rho5 protein which at least partially complemented the *rho5* deletion for both physiological read-outs. Besides providing the opportunity to study clinically important Rac1 variants in yeast, these results further underlined the importance of the yeast-specific insertion for proper spatiotemporal distribution and physiological function. It will be interesting to see if and how the Rho5 domains and their functions are conserved in other fungal Rho5 homologs, given their impact on hyphal morphologies with biotechnological applications [39].

## 4. Material and Methods

### 4.1. Strains and Growth Conditions

All yeast strains used are derived from the haploid strain HD56-5A. An isogenic diploid was previously obtained from this strain by introducing the *HO* gene on a plasmid and subsequent selection for plasmid loss [40,41]. HD56-5A is also closely related to the commonly employed CEN.PK series [42], as it was one of the parental strains used in the construction of these strains. Yeast cells were cultured in liquid media or on plates at 30 °C. Crossing, tetrad analyses and other genetic techniques followed standard procedures as described in reference [43]. Rich medium (YEPD) contained 1% yeast extract, 2% Bacto peptone (Difco Laboratories Inc., Detroit, MI, USA) and 2% glucose (all *w*/*v*). Synthetic media were prepared with Difco yeast nitrogen base (Difco Laboratories Inc., Detroit, MI, USA) with ammonium sulphate as described in reference [43], with the addition of amino acids and bases using a mixture provided by MP Biomedicals (Eschwege, Germany; CSM-His-Leu-Trp-Ura) supplemented histidine, leucine, tryptophan, and uracil, with omission of either histidine, leucine, or uracil, if required for selection of plasmid maintenance or when the respective wild-type alleles of the encoding genes were employed as deletion markers. Two percent glucose (*w*/*v*) was added as carbon source (SCD). Then, 200 mg/L of G418 were used for selection of the *kanMX* marker. Hydrogen peroxide was added at the concentrations indicated to induce oxidative stress.

To record growth curves, cells were pre-grown overnight in either rich or synthetic medium at 30 °C on a shaker set at 180 rpm. Cells were inoculated in fresh medium at an OD_600_ of 0.3 and grown again to mid-logarithmic phase. They were diluted to an OD_600_ of 0.1 in fresh medium and distributed in 100 µL aliquots into 96 well plates (Thermo Scientific, Bremen, Germany), with or without hydrogen peroxide as indicated. Growth was recorded in a Varioscan Lux plate reader (Thermo Scientific, Bremen, Germany) following the increase in OD_600_ by measuring each well once every 30 min for at least 18 h. Temperature was set at 30 °C and plates were moved with 1024 rpm with 5 s intervals, using the SkanIt Software 4.1 (Thermo Scientific, Bremen, Germany) for microplate readers, version 4.1.0.43. A factor of 3.546 was used for normalization with standard 1 cm cuvettes, as the path length in the cell culture was 0.282 cm. A minimum of two biological replicates (usually different segregants with the same mutant alleles obtained from tetrad analyses) and two technical replicates were recorded to generate each growth curve from mean values giving standard error bars.

Tetrad analyses were performed on YEPD plates using zymolyase 20T (MP Biomedicals) for digestion of the ascus walls and a Singer Instruments micromanipulator (Singer Instruments, Somerset, UK) as described in reference [15]. Plates were incubated for 3 days at 30 °C to allow for germination and growth of colonies. Images of the plates were then recorded by scanning with an Epson Perfection V500 Photo Scanner (Suwa, Nagano Prefecture, Japan). Brightness and contrast were adjusted for entire plates using Corel Photo Paint (Corel, Ottawa, ON, Canada). prior to selection of exemplary four tetrads each. A minimum of 18 tetrads were examined for each cross.

For manipulations in *E. coli*, strain DH5α was employed as described previously [44], with Luria broth media (1% *w*/*v* Bacto tryptone, 0.5% *w*/*v* yeast extract, 1% *w*/*v* sodium chloride) supplemented with either 50 mg/L ampicillin or 20 mg/L kanamycin, as required for plasmid selection.

### 4.2. Plasmid and Strain Constructions

Modifications at the original genetic loci were achieved by using one-step gene replacement techniques [45]. To avoid problems with copy numbers and plasmid stability, *RHO5* variants were tagged and used to substitute the gene at its original locus. In brief, cloned variants were equipped with the *SkHIS3* cassette from the Longtine collection [46] by in vivo recombination in their 3′-non-coding regions, flanked by sequences homologous to the native *RHO5* locus. From there, genomic sequences were replaced by homologous recombination selecting for histidine prototrophy. Mitochondria were visualized by a C-terminal Idp1-mCherry fusion encoded at the native *IDP1* locus, obtained by in vivo recombination with a PCR-generated cassette using *SkHIS3* as a selection marker. Sequences of all fusion constructs at their genetic loci are available upon request.

Trapping of GFP-Rho5 to mitochondria was achieved by fusion of the sequence encoding a GFP-binding monomeric antibody (GB; shortly referred to as “GFP-binder”), obtained by PCR from a template kindly provided by Roland Wedlich-Söldner [47], to the one encoding the transmembrane domain of the outer mitochondrial membrane protein Fis1 (GB-Fis1) in the yeast integrative plasmid YIplac128 [48] to obtain plasmid pLAO12. The *GB-FIS1* gene was expressed under the control of a constitutive yeast *PFK2* promoter tailored based on published promoter analyses [49], with 576 bp prior to the ATG translation start codon, which was previously shown to be suitable for heterologous expression of plant genes in yeast [50]. Intracellular distribution of GFP-Rho5 encoded on *CEN/ARS* vectors derived from YCplac33 or YCplac111 [48] was followed by fluorescence microscopy. Mutations at the 3′-end of the *RHO5* open reading frame were introduced by inverse PCR using appropriate oligonucleotides. For human *RAC1*, string-DNA synthesis was ordered from GeneArt (Thermofisher, Bremen, Germany) with the open reading frame optimized for *S. cerevisiae* codon usage and first cloned into pUK1921 for sequence verification [51]. Overexpression of a *RAC1-RHO5* hybrid construct obtained by in vivo recombination was achieved by placing the coding sequence under the control of the strong *TEF2* promoter from *S. cerevisiae* carried on YCplac111 [48]. A list of all plasmids employed is given in Table 2, and their sequences and details on their constructions are available upon request.

For assembly and alignments of sequences the Clone Manager 9 program (Scientific and Educational Software, Denver, CO, USA) was employed.

### 4.3. Fluorescence Microscopy

For standard microscopic examination cells were grown to early logarithmic phase in SCD medium. To apply oxidative stress, 4.4 mM hydrogen peroxide was added for 5–15 min prior to image acquisition throughout all localization studies, since these conditions were optimized previously [14]. Fluorescence microscopy was performed with a Zeiss Axioplan 2 (Carl Zeiss, Jena, Germany) microscope equipped with a 100× alpha-Plan Fluor objective (numerical aperture 1.45) and differential-interference contrast (DIC), as described previously [14]. Images were acquired using a Photometrics CoolSNAP HQ Camera (Roper Scientific, Tucson, AZ, USA). Fluorescence was excited with a SPECTRA X light engine (Lumencor, Beaverton, OR, USA). For colocalization studies, the microscope was additionally equipped with an image-splitter (DualView, Photometrics, Tucson, AZ, USA). The setup was controlled by the Metamorph v6.2 program (Universal Imaging Corporation, Downingtown, PA, USA). Images in the mCherry channel were obtained by 0.5 s exposures, and those in the GFP channel by 2 s exposures. Brightfield images were acquired as single planes using DIC. Scale bars were generated using Metamorphs scale image command. Huygens essential software (Scientific Volume Imaging, Hilversum, Netherlands) was used for deconvolution of the images. In order to visualize colocalization of signals from the mCherry- and the GFP-channels from the same cell, the processed images were overlaid using Metamorphs’ overlay images command.

A minimum of 100 cells (usually much more, with the total number “*n*” given in the figures) was examined for each strain and condition. Distributions of Rho5-GFP signals were judged to belong to one of three patterns, and the number of cells displaying association of the signal with the plasma membrane (pm), being either cytosolic or associated with internal structures (int), and those showing colocalization with the mitochondrial marker (mit) were determined. These numbers were divided by the total number of cells examined in each case and multiplied by 100 to calculate the percentages of cells showing the respective localization. Figures show only a few representative examples of the hundreds of images examined for each condition and strain.

## Figures and Tables

**Figure 1 ijms-20-05550-f001:**
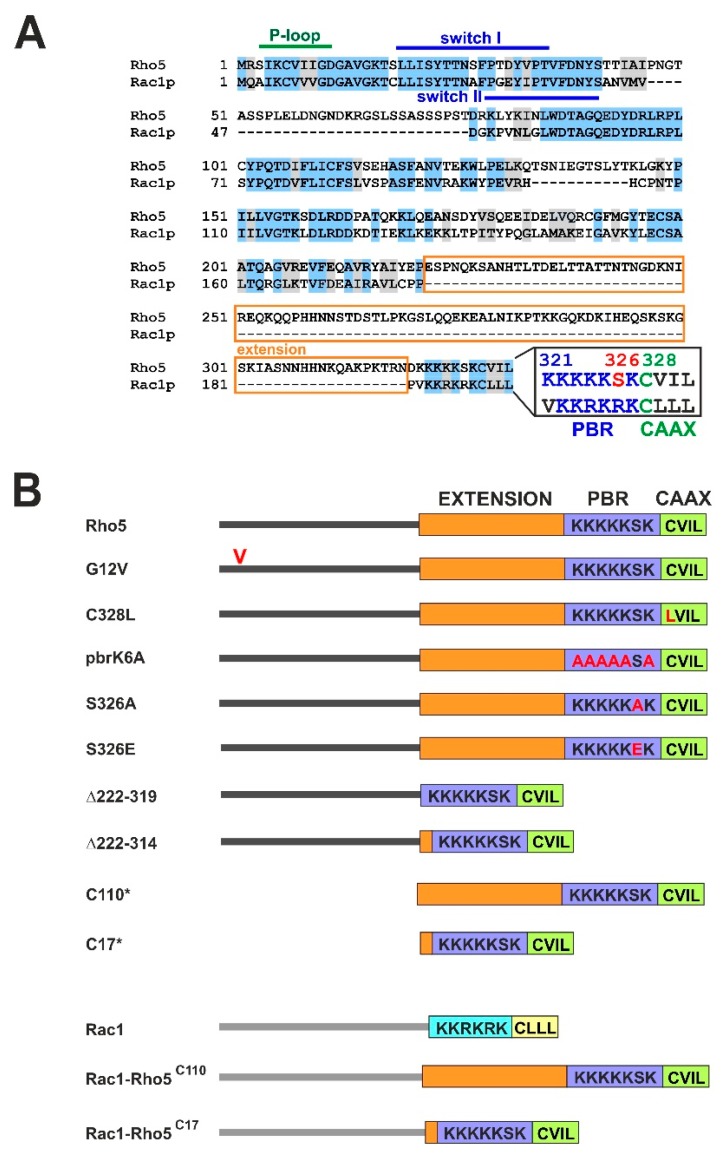
Comparison of yeast Rho5 with its human Rac1 homolog and mutants constructed. (**A**) Alignment of the primary sequences of Rho5 and Rac1. Identical amino acid residues are shaded in blue, conserved ones in grey. Domains relevant for the activation of the GTPases (P-loop, switch I and II regions), as well as the conserved sequences at the C-terminal end (PBR = polybasic region; CAAX = site of lipid modification, with a cysteine, two aliphatic and one variable terminal residues) are highlighted. The orange box designates a 98 amino acid extension specific for Rho5 of *Saccharomyces cerevisiae*. Accession numbers for the coding sequences are KZV08440 for *S. cerevisiae* Rho5, and CAB53579 for human Rac1. (**B**) Schematic representation of mutants used in this study. Designations of Rho5 and Rac1 variants used in the text are depicted at the left. Exchanges of amino acid residues are shown in red letters, including the valine near the N-terminus in the G12V mutant. Dark and light grey lines represent the catalytic domains of Rho5 and Rac1, respectively. Rectangles in orange designate the yeast-specific sequences of the 98 aa extension. The polybasic regions are highlighted in dark blue for Rho5 and light blue for Rac1, the CAAX motif in green for Rho5 and yellow for Rac1. For localization studies, all constructs were N-terminally tagged with GFP and expressed from *CEN/ARS* vectors. * C110 and C17 were exclusively used as N-terminal GFP fusions and not as untagged versions at the genomic locus.

**Figure 2 ijms-20-05550-f002:**
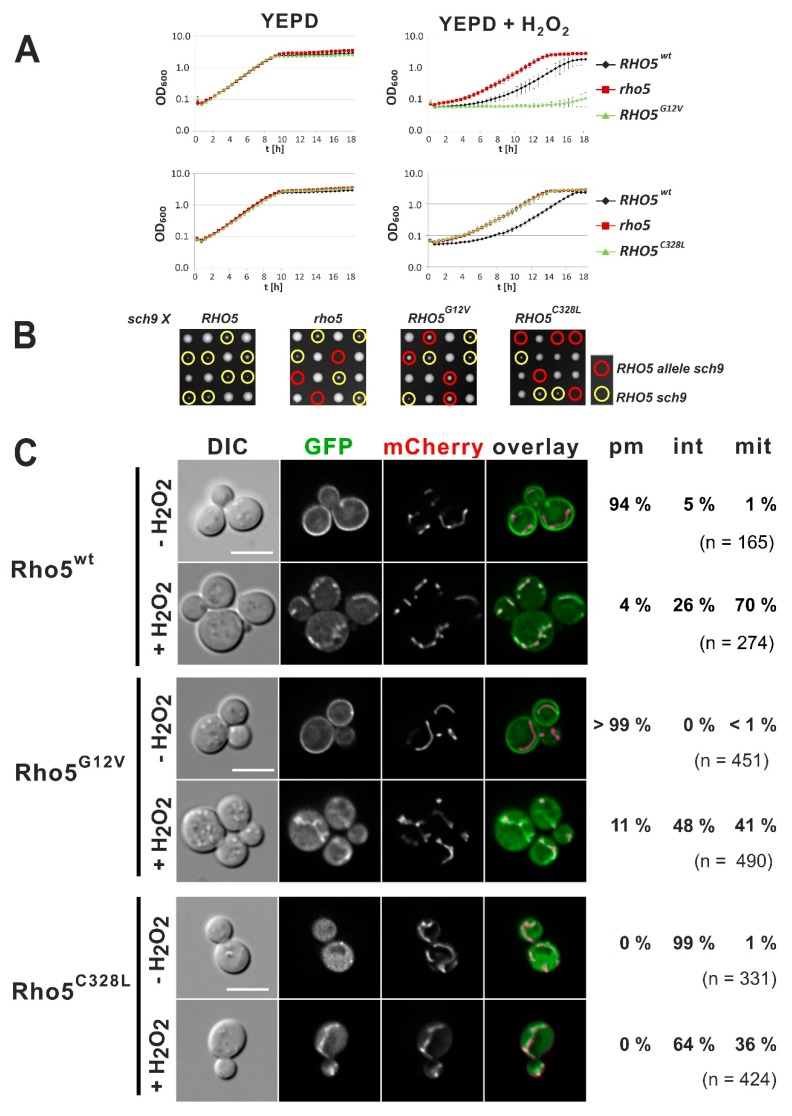
Growth and response to oxidative stress of Rho5^G12V^ and Rho5^C328L^ mutants. (**A**) Comparative growth behavior of strains with the indicated *RHO5* alleles. Growth curves were recorded in rich medium in the absence or presence of hydrogen peroxide as indicated. (**B**) Tetrad analyses of crosses from strains carrying the indicated *RHO5* allele to a *sch9* deletion strain performed on rich medium (YEPD). Note that *sch9* deletions grow more slowly and produce much smaller colonies than wild-type or *rho5* deletions, and that *rho5 sch9* double deletions are not viable. Yellow circles indicate tetrads carrying a wild-type *RHO5* allele and a *sch9* deletion, red circles mark the double mutants of *sch9* with the *RHO5* alleles indicated above each image. Parental strains crossed to HOD348-4D (*sch9*) from left to right were: HD56-RHwt (*RHO5*), HOD295-7D (*rho5*), HD56-R8 (*RHO5^G12V^*), and HD56-RH^C328L^ (*RHO5^C328L^*). See Table 1 for genotypes. (**C**) Life-cell fluorescence microscopy of a *rho5* deletion strain with a genomic *IDP1-mCherry* fusion as a mitochondrial marker (HCSO76-1A) expressing N-terminal GFP fusions of the Rho5 variants as indicated, each from a *CEN/ARS* vector. Cells were grown on synthetic media selecting for maintenance of the plasmid and either examined without further treatment (upper panels), or after addition of 4.4 mM hydrogen peroxide for less than 15 min (lower panels). The percentage of cells displaying association of the GFP signal with the plasma membrane (pm), being cytosolic or associated with internal structures (int), and those showing colocalization with the mitochondrial marker (mit) were determined for the total number of cells counted in each sample (*n*). Plasmids employed were pJJH1639 = GFP-Rho5; pLAO2 = GFP-Rho5^G12V^, and pCSO18 = Rho5^C328L^ (Table 2). Images in the mCherry channel were obtained by 0.5 s exposures, those in the GFP channel by 2 s exposures. The size bar in the upper left differential interference contrast (DIC) image represents 5 µm, applicable to all images in the panels of the same strain.

**Figure 3 ijms-20-05550-f003:**
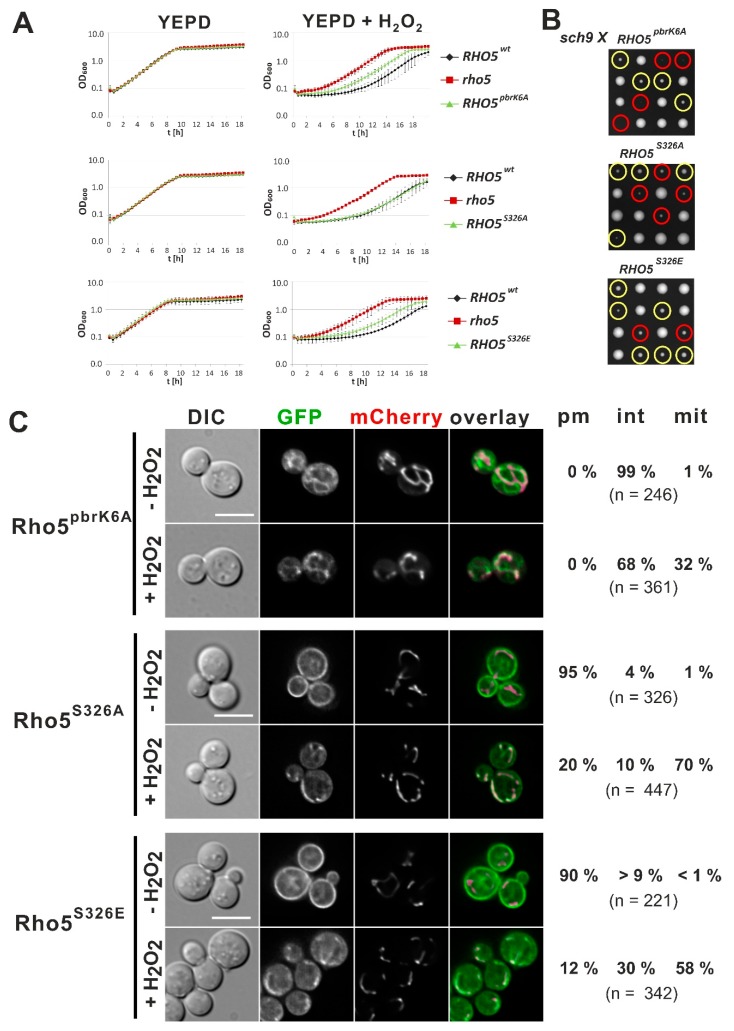
Growth and response to oxidative stress of mutants in the polybasic region of Rho5. (**A**) Growth curves were obtained in rich medium as described in materials and methods and the legend of Figure 2. (**B**) Tetrad analyses of crosses from strains carrying the indicated *RHO5* allele to a *sch9* deletion strain performed on YEPD. Parental strains employed in crosses to HOD348-4D (*sch9*) were HOD337-7D (*RHO5^pbrK6A^*), HD56-RH^S326A^ (*RHO5^S326A^*), and HD56-R7 (*RHO5^S326E^*). Yellow and red circles indicate segregants with a *RHO5 sch9* and a mutant *RHO5 sch9* constitution, respectively. (**C**) Life-cell fluorescence microscopy of a *rho5* deletion strain carrying *IDP1-mCherry* as a mitochondrial marker (HCSO76-1A) and the *RHO5* alleles indicated as N-terminal GFP fusions introduced on a *CEN/ARS* vector. Plasmids employed were pCSO35 = GFP-Rho5^pbrK6A^, pCSO19 = GFP-Rho5^S326A^, pCSO31 = GFP-Rho5^S326E^ (Table 2). Growth conditions, scales, and determination of distribution numbers were as described in the legend of Figure 2.

**Figure 4 ijms-20-05550-f004:**
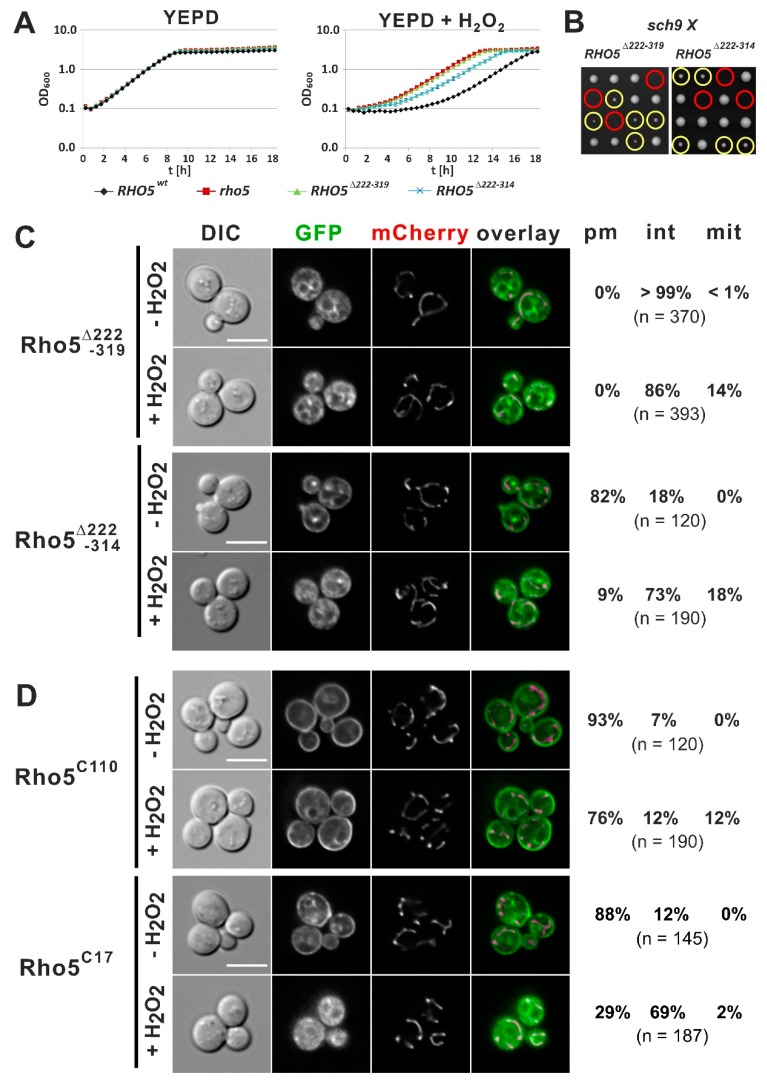
Growth and response to oxidative stress of a *rho5* mutant lacking the yeast-specific extension and analysis of its requirement for intracellular distribution. (**A**) Comparative growth behavior of strains carrying a wild-type *RHO5* allele, a *rho5* deletion, or the mutant allele lacking the extension (*RHO5^∆222^*^–*319*^) and the same allele with five more amino acids of the extension (*RHO5^∆222^*^–*314*^). (**B**) Tetrad analyses of crosses from strains carrying the indicated *RHO5* allele to a *sch9* deletion strain performed on YEPD. Parental strains employed were HD56-R2 (*RHO5^∆222^*^–*319*^) and HOD388-2D (*RHO5^∆222^*^–*314*^), each crossed with HOD348-4D (*sch9*). Yellow and red circles indicate segregants with a *RHO5 sch9* and a mutant *RHO5 sch9* constitution, respectively. (**C**,**D**) Life-cell fluorescence microscopy of a *rho5* deletion strain producing Idp1-mCherry as a mitochondrial marker (HCSO76-1A) and plasmid-encoded N-terminal GFP fusions of the Rho5 variants indicated. Plasmids introduced were pLAO5 (GFP-Rho5^∆222–319^), pJJH2616 (GFP-Rho5^∆222–314^), pJJH2562 (GFP-Rho5^C110^), pCSO94 (GFP-Rho5^C17^). Compare Table 2 for more information on plasmid features. Growth conditions, scales, and determination of distribution numbers were as described in the legend of Figure 2.

**Figure 5 ijms-20-05550-f005:**
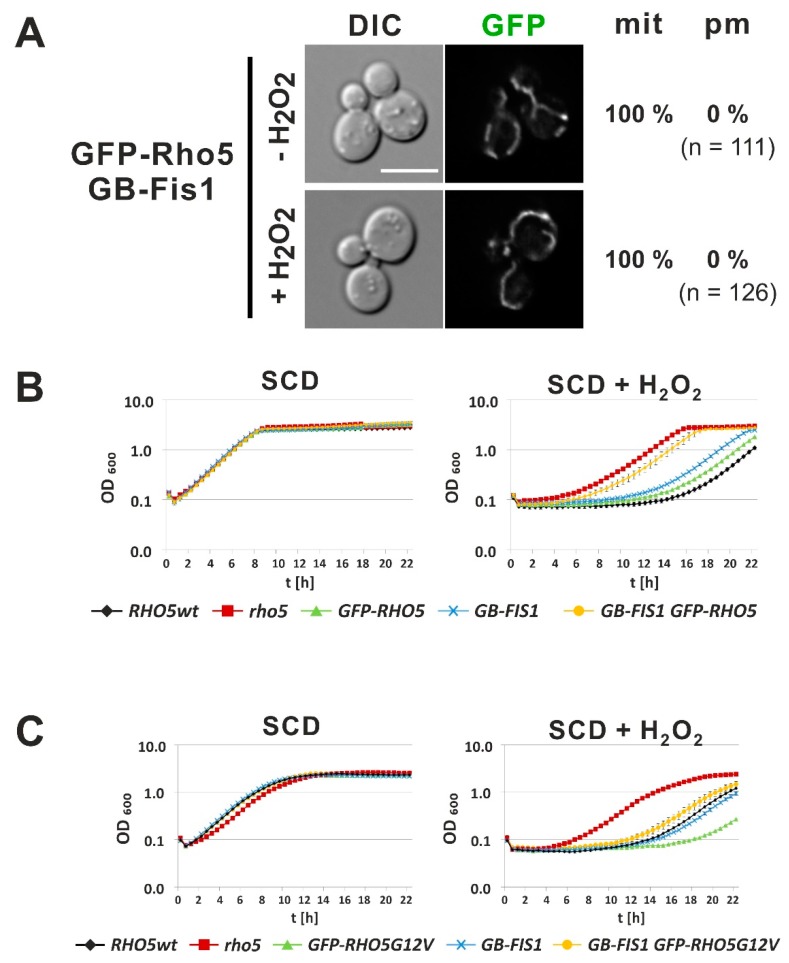
Physiological significance of trapping GFP-Rho5 to the mitochondrial surface. (**A**) Life-cell fluorescence microscopy of strain HOD370-2A, which encodes GFP-Rho5 at its genomic locus, as well as a GFP-binder fusion to the transmembrane domain of the outer mitochondrial protein Fis1, inserted into the *leu2-3,112* locus using an integrative plasmid. For detailed descriptions of growth conditions, scales and determination of distribution numbers see legend of Figure 2. (**B**) Comparative growth behavior of strains being wild-type for *RHO5* and *FIS1* (WT) or carrying different combinations of *GFP-RHO5* and *GB-FIS1*, as compared to a *rho5* deletion. Genotypes of strains employed are given in Table 1. Growth curves were recorded in synthetic medium in the absence or presence of hydrogen peroxide as indicated. (**C**) Comparative growth behavior of strains with an activated *GFP-RHO5^G12V^* allele. Growth curves were recorded in synthetic medium as described in (**B**).

**Figure 6 ijms-20-05550-f006:**
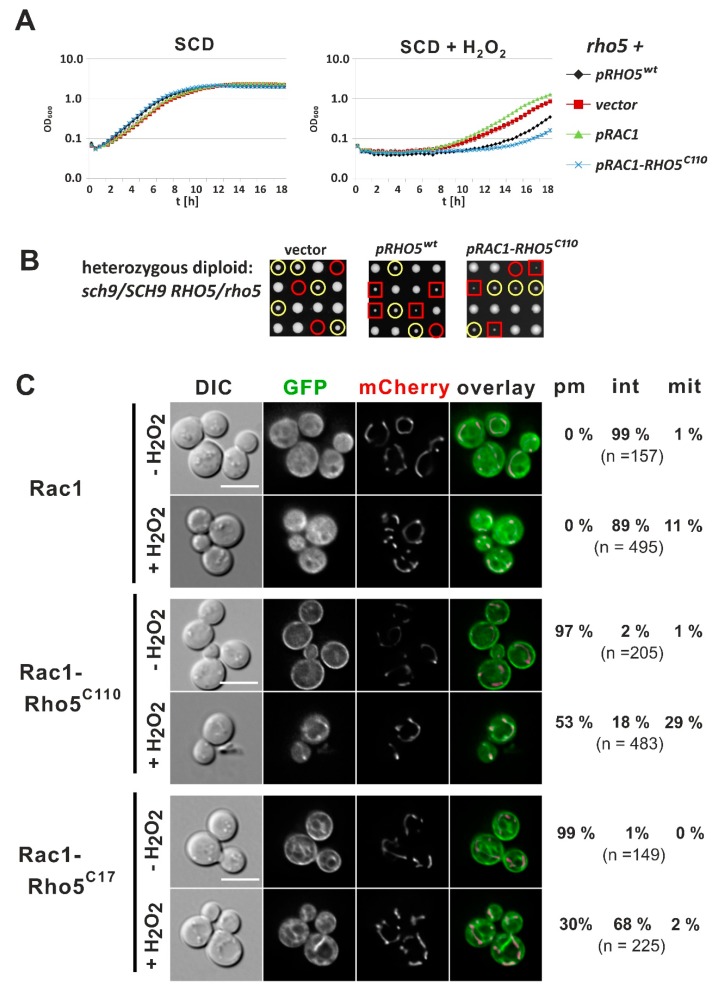
Growth and response to oxidative stress of yeast strains producing human Rac1 and its derivatives. (**A**) Comparative growth behavior of transformants of a *rho5* deletion (HAJ216-A) expressing the indicated GTPase variants from a centromeric vector. Plasmids introduced were YCplac111 (vector), pJJH1637 (*pRHO5^wt^*), pLAO4 (*pRAC1*), and pCSO85 (*pRAC1-RHO5^C110^*; see Table 2). (**B**) A heterozygous diploid obtained by crossing HAJ216-A (*rho5*) with HAJ195-A (*sch9*) was used as a recipient for the plasmids described in (**A**), with omission of pLAO4, and transformants were subjected to tetrad analyses. Yellow circles highlight segregants with a *sch9* deletion only, red circles non-viable segregants with the genotype *rho5 sch9*, and red squares designate double deletion segregants rescued by the plasmid-borne gene copy. (**C**) Life-cell fluorescence microscopy of a *rho5* deletion strain producing Idp1-mCherry as a mitochondrial marker (HCSO76-1A) and N-terminal GFP fusions of the Rac1 variants indicated, expressed from a *CEN/ARS* vector. Plasmids employed were pLAO6 (Rac1), pCSO89 (Rac1-Rho5^C110^), and pCSO100 (Rac1-Rho5^C17^). See Table 2 for more information on the plasmid features. For detailed descriptions of growth conditions, scales, and determination of distribution numbers see legend of Figure 2.

**Table 1 ijms-20-05550-t001:** Yeast strains used.

**Haploid Strains Employed**		
**Strain**	**Modifications Compared to HD56-5A Background ^1^** **(*MATα ura3-52 leu2-3,112 his3-11,15*)**	**Source**
HAJ216-A	*MATa rho5::kanMX*	[14]
HCSO76-1A	*MATa rho5::kanMX IDP1::mCherry-SkHIS3*	[15]
HCSO90wt	*MATa*	this work
HCSO90-3D	*RHO5^S326A^-SkHIS3*	this work
HCSO90-5A	*MATa RHO5^S326A^-SkHIS3*	this work
HCSO90-7B	*RHO5^S326A^-SkHIS3*	this work
HCSO91-1A	*RHO5-SkHIS3*	this work
HCSO91-3C	*MATa RHO5-SkHIS3*	this work
HCSO91-5A	*RHO5-SkHIS3*	this work
HCSO91-7A	*RHO5-SkHIS3*	this work
HCSO92-3A	*RHO5^S326A^-SkHIS3*	this work
HCSO94-1A	*RHO5^Δ222–319^*	this work
HCSO94-11B	*RHO5^Δ222–319^*	this work
HCSO95-1C	*MATa RHO5^S326E^-SkHIS3*	this work
HCSO95-5B	*RHO5^S326E^-SkHIS3*	this work
HCSO96-1A	*MATa RHO5^G12V^-SkHIS3*	this work
HCSO96-4C	*RHO5^G12V^-SkHIS3*	this work
HCSO97-1B	*MATa*	this work
HCSO97-3B	*RHO5^pbrK6A^-SkHIS3*	this work
HCSO97-5C	*MATa RHO5^pbrK6A^-SkHIS3*	this work
HCSO97-7C	*MATa*	this work
HCSO97-8A	*RHO5^pbrK6A^-SkHIS3*	this work
HCSO97-11B	*RHO5^pbrK6A^-SkHIS3*	this work
HCSO98-2B	*MATa rho5::SkHIS3*	this work
HCSO98-4A	*MATa rho5::SkHIS3*	this work
HCSO102-2B	*leu2-3,112::pLAO12 (GB-FIS1^TMD^-LEU2)*	this work
HD56-R2	*RHO5^Δ222–319^*	this work
HD56-R7	*RHO5^S328E^-SkHIS3*	this work
HD56-R8	*RHO5^G12V^-SkHIS3*	this work
HD56-RH^C328L^	*RHO5^C328L^-SkHIS3*	this work
HD56-RHO5^S326A^	*RHO5^S326A^-SkHIS3*	this work
HD56-RHwt	*RHO5-SkHIS3*	this work
HOD295-7D	*rho5::SkHIS3*	this work
HOD337-7D	*RHO5^pbrK6A^-SkHIS3*	this work
HOD348-4D	*MATa sch9::kanMX*	this work
HOD354-RHG	*MATa GFP-RHO5^G12V^-SkHIS3*	this work
HOD365-7B	*MATa GFP-RHO5-SkHIS3*	this work
HOD370-2A	*MATa GFP-RHO5-SkHIS3 leu2-3,112::pLAO12 (GB-FIS1^TMD^-LEU2)*	this work
HOD371-4A	*MATa GFP-RHO5-SkHIS3 leu2-3,112::pLAO12 (GB-FIS1^TMD^-LEU2)*	this work
HOD388-1C	*RHO5^Δ222–314^*	this work
HOD388-2D	*RHO5^Δ222–314^*	this work
HOD410-1A	*leu2-3,112::pLAO12 (GB-FIS1^TMD^-LEU2)*	this work
HOD410-1B	*MATa*	this work
HOD410-1C	*MATa GFP-RHO5^G12V^-SkHIS3*	this work
HOD410-1D	*GFP-RHO5^G12V^-SkHIS3 leu2-3,112::pLAO12 (GB-FIS1^TMD^-LEU2)*	this work
HOD410-3A	*leu2-3,112::pLAO12 (GB-FIS1^TMD^-LEU2)*	this work
HOD410-3B	*MATa GFP-RHO5^G12V^-SkHIS3*	this work
HOD410-3C	*GFP-RHO5^G12V^-SkHIS3 leu2-3,112::pLAO12 (GB-FIS1^TMD^-LEU2)*	this work
HOD410-3D	*MATa*	this work
**Diploid Strains Employed in Tetrad Analyses**		
**Strain**	**Modifications in DHD5 Background ^2^** **(*MATa/MATα ura3-52/ura3-52 leu2-3,112/leu2-3,112 his3-11,15/his3-11,15*)**	**Parental** **Strains**
DAJ138	*sch9::SkHIS3/SCH9 rho5::KanMX/RHO5*	from [15]
DCSO91	*sch9::kanMX/SCH9 RHO5-SkHIS3/RHO5*	HD56-RHwt HOD348-4D
DCSO92	*sch9::kanMX/SCH9 RHO5^S328A^-SkHIS3/RHO5*	HD56-RH^S326A^ HOD348-4D
DCSO93	*sch9::kanMX/SCH9 RHO5^C328L^-SkHIS3/RHO5*	HD56-RH^C328L^ HOD348-4D
DCSO94	*sch9::kanMX/SCH9 RHO5^Δ222–319^-SkHIS3/RHO5*	HD56-R2 HOD348-4D
DCSO95	*sch9::kanMX/SCH9 RHO5^S328E^-SkHIS3/RHO5*	HD56-R7 HOD348-4D
DCSO96	*sch9::kanMX/SCH9 RHO5^G12V^-SkHIS3/RHO5*	HD56-R8 HOD348-4D
DCSO97	*sch9::kanMX/SCH9 RHO5^pbrK6A^-SkHIS3/RHO5*	HOD337-7D HOD348-4D
DCSO98	*sch9::kanMX/SCH9 rho5::SkHIS3/RHO5*	HOD295-7D HOD348-4D
DHD5/dL + 5	*RHO5^Δ222–314^-SkHIS3/RHO5*	DHD5

^1^ HD56-5A is one of the parental strains of the CEN.PK series [42] and was first described in [40]. ^2^ DHD5 is an isogenic diploid derived from HD56-5A described in [41].

**Table 2 ijms-20-05550-t002:** Plasmids used in this study.

Name	Relevant Features	Source
pCSO18	*RHO5p GFP-RHO5^C328L^; LEU2; CEN/ARS; bla*	this work
pCSO19	*RHO5p GFP-RHO5^S326A^; LEU2; CEN/ARS; bla*	this work
pCSO31	*RHO5p GFP-RHO5^S326E^; LEU2; CEN/ARS; bla*	this work
pCSO35	*RHO5p GFP-RHO5^pbrK6A^; LEU2; CEN/ARS; bla*	this work
pCSO50	*PFK2p GFP-RHO5^C110^; LEU2; 2µ; bla*	this work
pCSO85	*TEF2p hsRAC1-RHO5; LEU2; CEN/ARS; bla*	this work
pCSO89	*RHO5p GFP-RAC1-RHO5^C110^; LEU2; CEN/ARS; bla*	this work
pCSO94	*RHO5p 2xGFP-RHO5^C17^; LEU2; CEN/ARS; bla*	this work
pCSO100	*RHO5p GFP-RAC1-RHO5^C17^; URA3; CEN/ARS; bla*	this work
pJJH1637	*RHO5p RHO5; LEU2; CEN/ARS; bla*	this work
pJJH1639	*RHO5p GFP-RHO5; LEU2; CEN/ARS; bla*	[15]
pJJH2562	*RHO5p GFP-RHO5^C110^; CEN/ARS; LEU2; bla*	this work
pJJH2616	*RHO5p GFP-RHO5^Δ222–314^; URA3; CEN/ARS; bla*	this work
pLAO2	*RHO5p GFP-RHO5^G12V^; LEU2; CEN/ARS; bla*	this work
pLAO4	*RHO5p hsRAC1; LEU2; CEN/ARS; bla*	this work
pLAO5	*RHO5p GFP-RHO5^Δ222–319^; URA3; CEN/ARS; bla*	this work
pLAO6	*RHO5p GFP-hsRAC1; URA3, CEN/ARS; bla*	this work
pLAO12	*PFK2p GFPbinder-FIS1^TMD^; LEU2; bla*	this work
YCplac33	*URA3; CEN/ARS; bla*	[48]
YCplac111	*LEU2; CEN/ARS; bla*	[48]
YIplac128	*LEU2; bla*	[48]

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
