# Peer review of "Analysis of Functional Domains in Rho5, the Yeast Homolog of Human Rac1 GTPase, in Oxidative Stress Response"

_ijms, 2019, doi:10.3390/ijms20225550_

Round 1

Reviewer 1 Report

Sterk et al, studied the conserved primary sequence in domain of C terminal region of Rho5 and their role in response to oxidative stress. They have analyzed in vivo function and intracellular distribution of Rho5. They show that Rho5 has a role in oxidative stress. The study is interesting, but the manuscript requires improvement in organization and clarity.

- The introduction presents the rationale for studying individual genes and its structure; however, the hypothesis for studying these agents as a composite should be presented.

- Authors used Hydrogen peroxide as stress, they mentioned the concentration once, were all their experiments at the same concentration. Did they try to use different concentration to observe the localization?

-  Will use Cadmium Chloride or Paraquat  as oxidative stress will have similar on Rho5 localization?

- The Graph in all the figures need to be edited for their Y axis, the number have comma instead of dot. i.e. 1,0. The optical density for different experiment shows a lot of variability for RHO5 wt, like Fig 2A both graphs with H2O2 have high variation on RHO5 wt. Please explain why this is so variable.

- There are no details how they have calculated the percentage of expression in PM, Int and Mit. Please add it fluorescent microscopy section.

Author Response

We thank the reviewer for the time invested and the constructive comments. We followed the suggestions given, as detailed for each point in the attached file.

Reviewer 1:

Sterk et al, studied the conserved primary sequence in domain of C terminal region of Rho5 and their role in response to oxidative stress. They have analyzed in vivo function and intracellular distribution of Rho5. They show that Rho5 has a role in oxidative stress. The study is interesting, but the manuscript requires improvement in organization and clarity.

- The introduction presents the rationale for studying individual genes and its structure; however, the hypothesis for studying these agents as a composite should be presented.

We apologize if the rationale was not sufficiently explained. While Rac1 has been studied extensively in mammalian cells and a variety of differential functions have been attributed to the hypervariable region at its C-terminal end, no similar studies are available on the yeast Rho5 homolog. Our aim was to shed some light on exactly this hypervariable region for yeast oxidative stress response, which in the case of Rho5 apparently includes the large 98 aa insertion. We added an explanatory sentence and rephrased the last part of the introduction, in order to clarify this point.

- Authors used Hydrogen peroxide as stress, they mentioned the concentration once, were all their experiments at the same concentration. Did they try to use different concentration to observe the localization?

Concentration and incubation times were tested in the previous work by Schmitz et al. (2015), which is cited. We now explained the conditions in M&M at beginning of chapter on fluorescence microscopy, to make this point clear.

- Will use Cadmium Chloride or Paraquat  as oxidative stress will have similar on Rho5 localization?

We expect that those compounds will have similar effects. While this is an interesting suggestion, we do not believe that it is essential to test these compounds for the purpose of this manuscript. Given the revision time of only 10 days, this was not practicable. However, we will keep the suggestion in mind for future works.

- The Graph in all the figures need to be edited for their Y axis, the number have comma instead of dot. i.e. 1,0. The optical density for different experiment shows a lot of variability for RHO5 wt, like Fig 2A both graphs with H2O2 have high variation on RHO5 wt. Please explain why this is so variable.

We apologize for the “Germanism” in axis numberings. Commata have been corrected for points and can be found in the revised manuscript (non-marked up, with accepted changes) and in the combined figure file.

Cells are very sensitive to small variations in cell density vs. H2O2 concentration and may take variable times to recover from lag phase. While we took every possible care in adjusting the cells to the same pregrowth conditions and diluting them to the same OD600 to start the experiment, ocassionally this still results in some variations indicated by the error bars. This is why biological replicas were tested in each individual experiments done in one plate under the same conditions. In fact, many more growth curves were obtained for most of the strains tested, which did not enter the manuscript, but supported the conlusions drawn. Since we do not interprete minor differences between different mutants lying within these error ranges, but rather the clearly different growth behaviours, we therefore are confident that the conclusions drawn are valid and supported by the growth curves shown.

- There are no details how they have calculated the percentage of expression in PM, Int and Mit. Please add it fluorescent microscopy section.

Details were given very concisely in the legend of Fig. 2. As suggested by the reviewer, we have added a more extensive explanation of how we calculated the localization percentages in the M&M section on fluo-microscopy.

Reviewer 2 Report

The authors have done a very nice work, examining how the hypervariable region at the carboxyl terminal region  of the small GTPase RHO5 controls in vivo function and its intracellular distribution of this GTPase. To do this, the authors have used an important collection of RHO 5 mutants and have elegantly demonstrated the relationship between this hypervariable region and the functionality of the GTPase, as well as its intracellular localization.

However, an issue that is not clear and should be explained: does GTPase RHO5 need to be activated in order to  translocate to the mitochondria? To clarify this point the authors could use the dominant form of RHO5 (RHO5 T20N).

As a minor point on page 11 last sentence, (Fig.6B) has to be replaced by (Fig.6C), the same on page 12 second line.

Author Response

We thank the reviewer for the time invested and the interesting and useful suggestions. We followed the advise.

Reviewer 2:

The authors have done a very nice work, examining how the hypervariable region at the carboxyl terminal region  of the small GTPase RHO5 controls in vivo function and its intracellular distribution of this GTPase. To do this, the authors have used an important collection of RHO 5 mutants and have elegantly demonstrated the relationship between this hypervariable region and the functionality of the GTPase, as well as its intracellular localization.

However, an issue that is not clear and should be explained: does GTPase RHO5 need to be activated in order to  translocate to the mitochondria? To clarify this point the authors could use the dominant form of RHO5 (RHO5 T20N).

We have included localization data on the Rho5-T17N mutant in supplementary Fig. S1, as suggested. And no, activation of Rho5 is not required for translocation, since this is mediated by interaction with the dimeric GEF Dck1/Lmo1, which presumably is not affected substantially by an inactive conformation of the GTPase. After all, the GEF interacts with the inactive form in order to activate it. Reference to Fig. S1 has been inserted at the end of the first results chapter and the data are briefly discussed in conjunction with those for the activated Rho5-G12V variant in the discussion.

Since time constraints in the revision process did not allow us to construct the genomic insertion of the RHO5-T17N allele and test for the phenotypic consequences of having the untagged mutant protein, as was done for the other mutants in the manuscript, we decided to add the localization studies suggested by the reviewer only as a supplementary figure, rather than a regular addition to the manuscript.

As a minor point on page 11 last sentence, (Fig.6B) has to be replaced by (Fig.6C), the same on page 12 second line.

We thank the reviewer for spotting these errors, apologize, and of course corrected them.

Round 2

Reviewer 2 Report

The answers issued by the authors are satisfactory, so I consider the work suitable for publication in IMJS.